

# Postoperative hyperlactatemia and serum lactate level trends among heart transplant recipients

Anna Kędziora[1], Karol Wierzbicki[1], Jacek Piątek[1], Hubert Hymczak[2], Izabela Górkiewicz-Kot[1], Irena Milaniak[1,3], Paulina Tomsia[1], Dorota Sobczyk[4], Rafal Drwila[2] and Boguslaw Kapelak[1]

[1] Department of Cardiovascular Surgery and Transplantology, John Paul II Hospital, Institute of Cardiology, Jagiellonian University Medical College, Krakow, Poland
[2] Department of Anesthesiology and Intensive Care, John Paul II Hospital, Krakow, Poland
[3] Wydział Lekarski i Nauk o Zdrowiu, Krakowska Akademia im. Andrzeja Frycza-Modrzewskiego, Kraków, Polska
[4] Department of Cardiac and Vascular Diseases, John Paul II Hospital Institute of Cardiology, Jagiellonian University Medical College, Krakow, Poland

Corresponding author
Anna Kędziora,
anna.kedziora@doctoral.uj.edu.pl,
anna.kedziora.mail@gmail.com

## ABSTRACT

**Background**. Advanced heart failure (HF), that affects 10% of the HF population, is associated with high mortality rate, meeting 50% at 1-year from diagnosis. For these individuals, heart transplantation (HTX) remains the ultimate and the gold-standard treatment option. Serum lactate level measurements has been proven useful for determining the outcome following other cardiac surgeries and among critically ill patients. Increased serum lactate levels are expected following HTX; however, no detailed analysis has been yet performed in this population. The research aims to estimate the prevalence of hyperlactatemia and describe early postoperative serum lactate level trends among heart transplant recipients.

**Materials and Methods**. Forty-six consecutive patients, who underwent HTX between 2010 and 2015, were enrolled into the retrospective analysis. Serum lactate level measurements within first 48 hours post-HTX were obtained every 6 hours from routinely conducted arterial blood gas analyses. The threshold for hyperlactatemia was considered at >1.6 mmol/L, according to upper limit of normal, based on internal laboratory standardization. The highest observed measurement within the observation, regardless of the time point of observation was determined for each patient individually and was appointed as *Peak Value*.

**Results**. Consecutively measured serum lactate levels differed in time ($p = 0.000$), with the initial increase and subsequent decrease of the values (4.3 vs. 1.9 mmol/l; $p = 0.000$). The increase from the baseline level to the *Peak Value* was statistically significant (4.3 vs. 7.0 mmol/l; $p = 0.000$). Various serum lactate level trends were identified, with one or more hyperlactatemia episodes. Eventually, 50% of the individuals had normal serum lactate levels at the end of the study, and hyperlactatemia was observed in the other half.

**Conclusions**. Throughout the observation, all of the patients experienced at least one episode of hyperlactatemia, with the median *Peak Value* of 7.0 (4.5–8.4) mmol/L. Various serum lactate level trends can be identified in post-HTX patients. Further research is required to determine the clinical usefulness of newly reported serum lactate level trends among heart transplant recipients.

# INTRODUCTION

Heart failure (HF), with the current prevalence of 2.5%, is a major public health problem associated with significant hospital admission rates, mortality, and health care costs (*Benjamin et al., 2017*). The age-adjusted rate for heart failure-related deaths has been steadily increasing since 2012, reaching 84.0 deaths per 100,000 standard population in 2014 (*Ni & Xu, 2015*). Advanced HF, that affects 10% of the HF patients, is associated with even higher mortality rate, meeting 50% at 1-year from diagnosis (*Lund, 2018*). For these individuals, heart transplantation (HTX) remains the ultimate and the gold-standard treatment option (*Ponikowski et al., 2016*). Nevertheless, despite the total of approximately 5,000 HTX performed worldwide annually (*Khush et al., 2018*), there is still a gap in scientific evidence in terms of early postoperative management for heart transplant recipients. The immunosuppression, anti-infective prophylaxis, and pulmonary hypertension treatment have been widely discussed in this group of patients (*Costanzo et al., 2010*), however, only few postoperative biochemical measurements have been thoroughly analyzed so far (*Wierzbicki et al., 2014*; *Madershahian et al., 2008*).

Under basal conditions, the lactate is produced by all tissues and reutilized at a nearly constant rate by the liver. Normally, lactate concentration in the extracellular fluid is about one mmol/l. The diagnostic lower and upper limits of normal slightly vary (from 0.3 to 2.0 mmol/l) according to the specimen (arterial or venous blood) and method used (*Toffaletti, 1991*). The overproduction and, therefore, increased levels can be observed during tissue oxygen deficiency as a result of the circulatory shock and perfusion failure (*Adeva-Andany et al., 2014*). The potential usefulness of lactate measurements as an indicator of the shock severity has been has been first evaluated in 1964 (*Broder & Weil, 1964*). Since then, numerous studies have reported data for measurements taken in various acute clinical settings and, throughout time, cut-off values for therapeutic management have evolved (*Zhou et al., 2017*).

Similarly, hyperlactatemia assessment has been proven useful in determining the outcome in some cardiosurgical procedures, as it reflects the problem of postoperative hypoperfusion (*Hajjar et al., 2013*; *Andersen et al., 2015*; *Lopez-Delgado et al., 2015*). Following HTX, increased serum lactate levels are expected, as prolonged cardiopulmonary bypass (CPB) time and compromised hemodynamic function within early postoperative hours are commonly observed. However, postoperative serum lactate level trends and clinical usefulness of lactate measurements have never been analyzed before among post-HTX patients. Moreover, currently available guidelines for the care of heart transplant recipients are mostly based on the consensus of opinion of the experts (level of evidence C) (*Costanzo et al., 2010*), which indicates the gap in scientific evidence.

Table 1 **Baseline characteristics.** Data shown as mean ± SD or as median (IQR), or number (percentage).

| Variable | Analyzed population; $n = 46$ |
|---|---|
| Age, years | $48.7 \pm 11.7$ |
| Male sex, $n$(%) | 41 (89.1) |
| Dilated cardiomyopathy, $n$(%) | 34 (73.9) |
| Ischemic cardiomyopathy, $n$(%) | 12 (26.1) |
| Dyslipidemia, $n$(%) | 16 (34.8) |
| Hypertension, $n$(%) | 21 (45.7) |
| Diabetes, $n$(%) | 12 (26.1) |

Therefore, the presented research aims to estimate the actual prevalence of hyperlactatemia and describe early postoperative serum lactate level trends among heart transplant recipients.

## MATERIAL AND METHODS

Forty-six consecutive patients, who underwent HTX in the Department of Cardiovascular Surgery and Transplantology between 2010 and 2015, were enrolled into the retrospective analysis. Serum lactate level measurements within first 48 h post-HTX were obtained from arterial blood gas (ABG) analyses, that were routinely conducted every 6 h. The mean age of the study group was $48.7 \pm 11.7$ years. The majority of the patients were males (89.1%) and were qualified for HTX due to dilated cardiomyopathy (73.9%) (Table 1).

Consecutive serum lactate level measurements, that were taken every 6 h, were appointed as *Lac*, with the following number as the time point (hour) of observation. The baseline measurement was defined as the observation taken at Intensive Care Unit (ICU) admission (*Lac 0*). The end of the study measurement was defined as the observation taken at the 48th hour post-HTX (*Lac 48*). The highest observed measurement within the observation, regardless of the time point of observation was determined for each patient individually and was presented as the *Peak Value*. The threshold for hyperlactatemia (>1.6 mmol/L) was defined by the upper limit value of the normal range for serum lactate concentration (0.5–1.6 mmol/l), based on internal laboratory standardization for ABG analysis. ABG analysis was performed with an automated electrode analyzer (*Arterial Blood Gases-Clinical Methods-NCBI Bookshelf, 0000*).

The study obtained the Jagiellonian University Bioethics Committee approval (122.6120.74.2017). According to the approval, verbal consent was obtained from all living patients.

### Statistical analysis

Statistical analysis was performed with IBM® SPSS® Statistics 25. Normal distribution was tested using the Shapiro–Wilk test. Continuous variables are presented as means and standard deviation (±SD) or medians and interquartile ranges (IQR). For categorical variables, numbers and proportions are reported. Non-parametric tests for independent
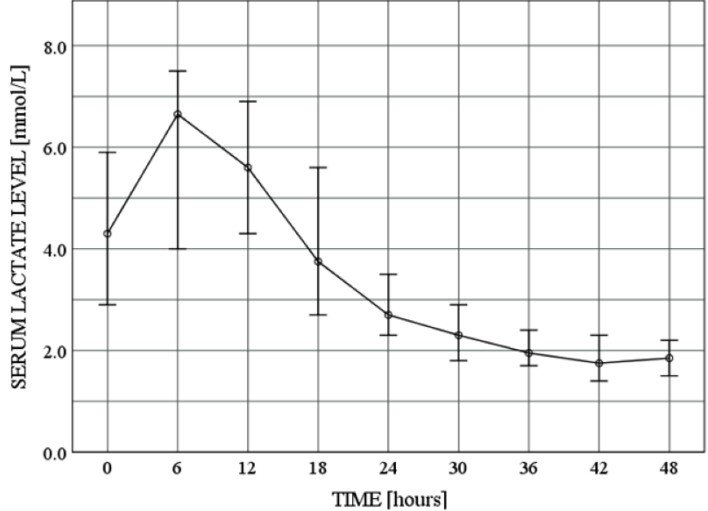

MEDIAN SERUM LACTATE LEVELS WITHIN 48 HOURS OBSERVATION

| OBSERVATION TIME POINT | SERUM LACTATE LEVEL [mmol/L] |
|---|---|
| Lac 0 | 4.3 (2.3 – 6.4) |
| Lac 6 | 6.7 (3.3 – 8.0) |
| Lac 12 | 5.6 (3.7 – 7.6) |
| Lac 18 | 3.8 (2.3 – 6.1) |
| Lac 24 | 2.7 (1.9 – 4.0) |
| Lac 30 | 2.3 (1.6 – 3.1) |
| Lac 36 | 2.0 (1.6 – 2.5) |
| Lac 42 | 1.8 (1.3 – 2.4) |
| Lac 48 | 1.9 (1.3 – 2.3) |
| Peak Value | 7.0 (4.5 – 8.4) |

Error Bars: 95% CI

**Figure 1 Serum lactate level trends within 48 hours observation.** Serum lactate level distributions differ in time: $\chi^2(2) = 169.1$; $p = 0.000$; Friedman test. Serum lactate level medians differ between baseline (Lac 0) and end of the study (Lac 48): $p = 0.000$; Wilcoxon signed-rank test. Serum lactate level medians differ between baseline (Lac 0) and peak time point (*Peak Value*): $p = 0.000$; Wilcoxon signed-rank test Data for serum lactate levels are presented as medians (IQR). Lac X—consecutive serum lactate measurements with the following number as the time point (hour) of observation. *Peak Value*—the highest observed serum lactate level within the observation, regardless of the time point.

(Kruskal–Wallis test) or dependent samples (Wilcoxon signed-rank test, Friedman test), where appropriate, were calculated to determine the differences between observations.

## RESULTS

The values of the consecutive serum lactate level measurements differed in time ($p = 0.000$; Friedman test) and initial increase with subsequent decrease throughout the study was observed. The increase from the baseline level (*Lac 0*) to the highest observed value (*Peak Value*) was statistically significant ($p = 0.000$; Wilcoxon signed-rank test), and similarly was the decrease throughout the observation (from *Lac 0* to *Lac 48*; $p = 0.000$; Wilcoxon signed-rank test) (Fig. 1).

In the detailed analysis, various serum lactate level trends were identified. In the majority of the patients (63%) serum lactate level normalization was achieved, however, one-third (33%) of the study group experienced sequent episode of hyperlactatemia after first normalization. Eventually, 50% of the individuals had normal serum lactate levels at the end of the study (*Lac 48*), and hyperlactatemia was observed in the other half (Fig. 2).

Further calculations determined the association between the baseline serum lactate level (*Lac 0* as *normal* or *hyperlactatemia* or *peak lactates*) and the highest observed serum lactate level (*Peak Value*) ($p = 0.06$; Kruskal–Wallis test; Fig. 3). No such association was found for the end of the study serum lactate level (*Lac 48* as *normal* or *hyperlactatemia*) ($p = 0.263$; Kruskal–Wallis test; Fig. 4).

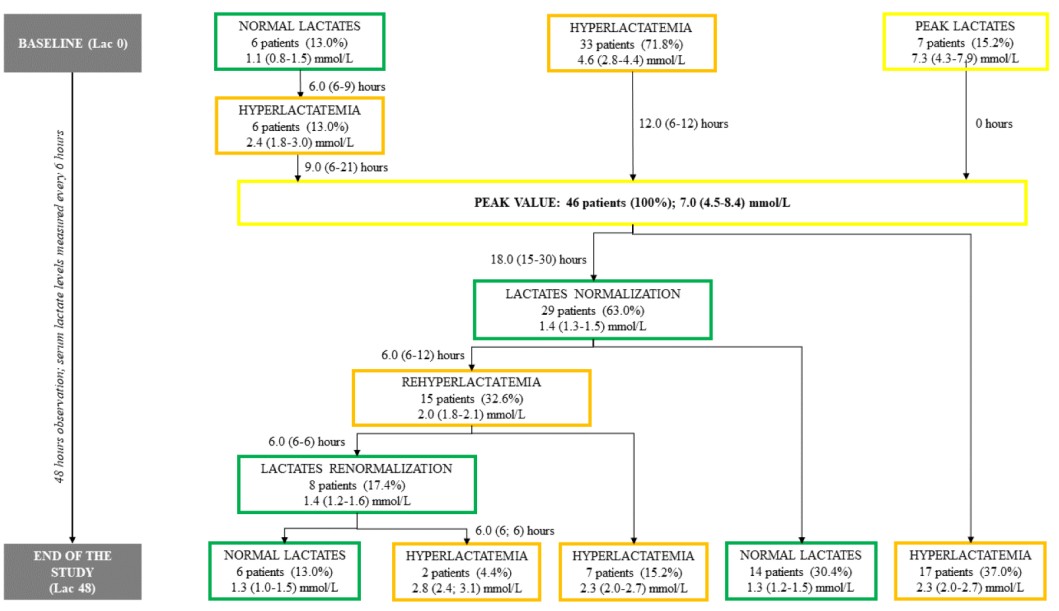

**Figure 2  Study flow chart.** Data for time intervals and serum lactate levels are presented as medians (IQR). The number of patients at each point is followed by the % based on the total number of patients in the study. Lac X –consecutive serum lactate measurements with the following number as the time point (hour) of observation. *Peak Value*—the highest observed serum lactate level within the observation, regardless of the time point.

## DISCUSSION

Under anaerobic conditions, cellular metabolism converts pyruvate to lactate. Therefore, serum lactate levels relate to the oxygen debt and correspond with the severity of tissue hypoperfusion (*Adeva-Andany et al., 2014*; *Hoshino, Kinoshita & Ono, 2018*). The normal plasma lactate concentration for ABG is reported from 0.3 to 1.6 mmol/L (*Broder & Weil, 1964*), whereas lactic acidosis is generally defined with a threshold of greater than four mmol/L, even in the absence of overt acidemia. Type A lactic acidosis, that is considered to be a medical emergency and is associated with poor prognosis when left untreated, is usually caused by significant tissue hypoperfusion (*Andersen et al., 2013*).

Hyperlactatemia is a common phenomenon following cardiac surgeries, as it can result from not only hypoxic but also nonhypoxic causes such as drug therapy, cardioplegia, hypothermia, and CPB. Previous studies among cardiosurgical patients, comparing survivors vs. non-survivors, have indicated that that higher lactate levels upon ICU admission (2–3 mmol/l), or within early postoperative hours (3–4 mmol/l), are associated with increased perioperative risk and prolonged hospital stay (*Hajjar et al., 2013*; *Lopez-Delgado et al., 2015*; *Andersen et al., 2015*).

Nevertheless, to date only two studies, concerning the issue of postoperative hyperlactatemia among heart transplant recipients have been published. Both of the researches were single center studies that evaluated solely whether the patient has reached the threshold, set by the authors, at any time point throughout the observation. The
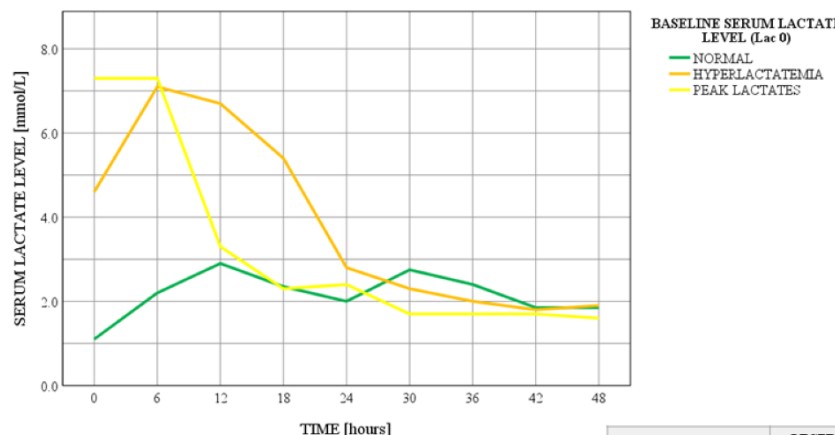

| BASELINE | OBSERVATION TIME POINT | SERUM LACTATE LEVEL [mmol/L] |
|---|---|---|
| NORMAL | Lac 0 | 1.1 (0.8 – 1.5) |
| | Lac 48 | 1.9 (1.3 – 2.5) |
| | Peak Value | 3.8 (2.9 – 4.5) |
| HYPERLACTATEMIA | Lac 0 | 4.6 (2.8 – 6.4) |
| | Lac 48 | 1.9 (1.4 – 2.5) |
| | Peak Value | 7.6 (5.3 – 9.0) |
| PEAK LACTATES | Lac 0 | 7.3 (4.3 – 7.9) |
| | Lac 48 | 1.6 (1.3 – 2.2) |
| | Peak Value | 7.3 (4.3 – 7.9) |

**Figure 3  Serum lactate level trends within 48 hours observation by baseline serum lactate level.** Baseline (Lac 0) serum lactate level distributions differ among the groups: $p = 0.000$; Kruskal–Wallis test. Peak (*Peak Value*) serum lactate level distributions differ among the groups: $p = 0.06$; Kruskal–Wallis test. Data for serum lactate levels are presented as medians (IQR). Lac X—consecutive serum lactate measurements with the following number as the time point (hour) of observation. *Peak Value*—the highest observed serum lactate level within the observation, regardless of the time point.

extreme hyperlactatemia was defined by Hsu et al. as the serum lactate level greater than 15 mmol/L and occurred in 3 patients at the ICU admission and in 9 others within early postoperative hours (12/58 patients). In spite of the further serum lactate level decrease below four mmol/L, one-third of the analyzed hyperlactatemia subgroup deceased within the hospitalization, and the others within nearly 5 years post-HTX (*Hsu et al., 2015*). In another study, the cut-off level of four mmol/L was set according to the guidelines for severe sepsis and septic shock treatment. Based on this estimation, elevated serum lactate levels were observed among 59.2% of post-HTX patients, with higher occurrence of postoperative pulmonary complications within this subgroup. As no in-hospital deaths were reported, no testing was performed to determine the impact on early mortality (*Hoshino, Kinoshita & Ono, 2018*). Nevertheless, the detailed exploration of the cut-off value, appointed by the upper limit of normal for serum lactate concentration, has never been presented in post-HTX patients before.

In the presented research, hyperlactatemia was considered with the of >1.6 mmol/l, same as it is for other cardiac surgeries and non-surgical patients in our center. Despite

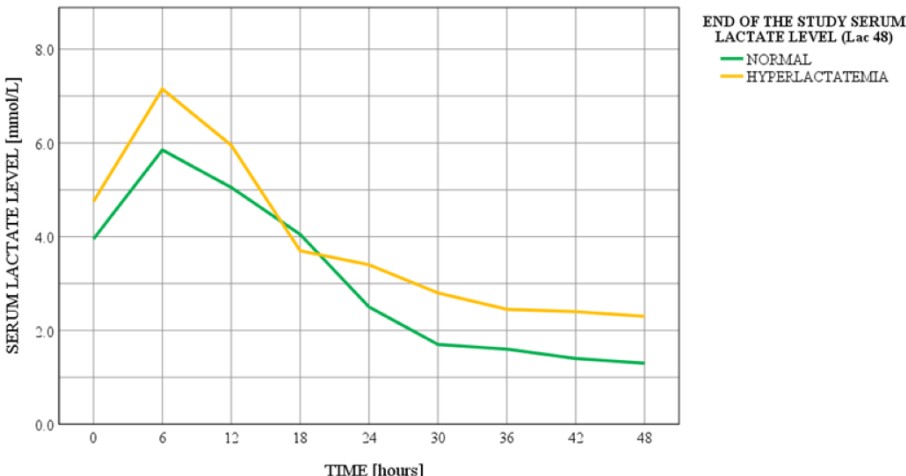

| END OF THE STUDY | OBSERVATION TIME POINT | SERUM LACTATE LEVEL [mmol/L] |
|---|---|---|
| NORMAL | Lac 0 | 4.0 (2.2 – 6.2) |
| | Lac 48 | 1.3 (1.1 – 1.5) |
| | Peak Value | 6.2 (3.9 – 8.3) |
| HYPERLACTATEMIA | Lac 0 | 4.8 (2.5 – 6.6) |
| | Lac 48 | 2.3 (2.0 – 2.7) |
| | Peak Value | 7.6 (4.5 – 9.0) |

**Figure 4  Serum lactate level trends within 48 hours observation by end of the study serum lactate level.** End of the study (Lac 48) serum lactate level distributions differ among the groups: $p = 0.000$; Kruskal–Wallis test. Data for serum lactate levels are presented as medians (IQR). Lac X –consecutive serum lactate measurements with the following number as the time point (hour) of observation. *Peak Value*—the highest observed serum lactate level within the observation, regardless of the time point.

the initially emerging pattern with early peak of the lactate values, that was similar to other post-HTX lactate analyses (*Hsu et al., 2015*; *Hoshino, Kinoshita & Ono, 2018*), more detailed evaluation revealed various serum lactate level trends. The interpretation of the initial results indicates the increase to the highest serum lactate level within first 6 postoperative hours with subsequent gradual decrease (Fig. 1). However, close case-by-case investigation provided more rigorous outcome, showing differences in the number of hyperlactatemia episodes and time intervals in between (Fig. 2).

Serum lactate level trends significantly differed between subgroups assigned based on the baseline serum lactate level (*Lac 0* as *normal* or *hyperlactatemia* or *peak lactates*). The highest observed measurement (*Peak Value*) varied between the subgroups. Although the lowest *Peak Value* was seen in the *normal* group, two serum lactate level spikes can be identified (12th and 30th hour of observation). Based on the metadata, in half of these patients *Peak Value* was noted at 12th hour post-surgery, and in the other half later within observation. Moreover, one-third of these individuals actually experienced sequent hyperlactatemia episode (Fig. 3). On the other hand, patients who entered the study with their highest serum lactate levels (*Lac 0* as *peak lactates*), completed the analysis with

lowest values after a considerable curve slope within first 12 h of observation (Fig. 3). Similar curve slope difference was noted among the subgroups assigned by the 48th hour measurement (*Lac 48* as *normal* or *hyperlactatemia*), however, this trend changed at 18th hour of observation, resulting in elevated serum lactate levels at the end of the study in the *hyperlactatemia* subgroup (Fig. 4).

The planned extension of this study includes the search for pre- and intraoperative factors determining various serum lactate level trends, as well as for the association between various serum lactate level trends and the postoperative management and course, with the special focus on the hemodynamic function of the transplanted heart.

## CONCLUSIONS

Throughout the observation, all of the patients experienced at least one episode of hyperlactatemia, with the median peak value of 7.0 (4.5–8.4) mmol/L. Various serum lactate level trends can be identified in post-HTX patients, with the greatest differences in terms of the ICU admission measurement, and the highest observed value. Further research is required to determine the clinical usefulness of newly reported serum lactate level trends among heart transplant recipients.

### Funding
The study was funded by the grant from Jagiellonian University Medical College (K/ZDS/007226). The funders had no role in study design, data collection and analysis, decision to publish, or preparation of the manuscript.

### Grant Disclosures
The following grant information was disclosed by the authors:
Jagiellonian University Medical College: K/ZDS/007226.

### Competing Interests
The authors declare there are no competing interests.

### Author Contributions
- Anna Kędziora, Karol Wierzbicki and Jacek Piątek conceived and designed the experiments, performed the experiments, analyzed the data, prepared figures and/or tables, authored or reviewed drafts of the paper, and approved the final draft.
- Hubert Hymczak analyzed the data, authored or reviewed drafts of the paper, and approved the final draft.
- Izabela Górkiewicz-Kot, Dorota Sobczyk and Rafal Drwila performed the experiments, authored or reviewed drafts of the paper, and approved the final draft.
- Irena Milaniak performed the experiments, prepared figures and/or tables, and approved the final draft.

- Paulina Tomsia performed the experiments, analyzed the data, prepared figures and/or tables, and approved the final draft.
- Boguslaw Kapelak conceived and designed the experiments, authored or reviewed drafts of the paper, and approved the final draft.

## Ethics

The following information was supplied relating to ethical approvals (i.e., approving body and any reference numbers):

The study obtained the Jagiellonian University Bioethics Committee's approval (122.6120.74.2017).

## Data Availability

The raw data is available as a Supplementary File.

## Supplemental Information

Supplemental information for this article can be found online at http://dx.doi.org/10.7717/peerj.8334#supplemental-information.

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
