# Peer review of "Postoperative hyperlactatemia and serum lactate level trends among heart transplant recipients"

_PeerJ, doi:10.7717/peerj.8334_

## Round 0.1 · original submission · Minor Revisions

All critiques of both reviewers need to be addressed and the manuscript should be revised accordingly.

Reviewer 1 ·

Basic reporting

-English needs to be improved at various places including the Abstract section for a clearer understanding by the audience. Last half of the background section is hard to understand.

-Typos are also abundant. For example in the materials and method section-Serum lactate level measurements within first 48 hours post-HTX were obtained every 6 six hours from routinely conducted arterial blood gas analyses- 6 six, please remove extra 6.

-Line 54 has a typo :from Lac 0 to Lac 48; p= p=0.000; Wilcoxon signed-rank test. Please remove extra p=.

-Line 51 is very hard to understand.

-Change from passive to active tense is very frequent in the materials and method section which should be avoided.

Experimental design

I commend the authors for evaluating the usefulness of serum lactate levels in their study as a measure of post-HTX operative care. However, I would like the authors to consider the following suggestions-

1. It would be a more comprehensive study if normal serum lactate levels were calculated from patients without HTX by the authors instead of using published values from literature. Serum lactate trends in normal individuals will also contribute greatly to the study.
2. More information on 'internal lab standardization' will help readers understand the threshold values selected by the authors
3. It would add greatly to the study if the authors made efforts to describe alternate measures of evaluation of post HTX outcome

Validity of the findings

none

Additional comments

1. The resolution of figures needs improvement
2. Figure Legends needs improvement

Reviewer 2 ·

Basic reporting

This manuscript reports a detailed study of serum lactate levels in post-heart transplant patients which can be extremely useful to determine the possible outcome of patients postoperatively.

Experimental design

In the study, the authors did not present the research question clear enough. I think they should include more background and details from the previous reports for better understanding to broad interest readers.

Validity of the findings

No comment

Additional comments

I have raised a few issues which are pointed out below:

1. Please re-write the abstract section especially the background and results section. In the background section, the research problem is not presented clearly.
2. I suggest to re-write the introduction section of the manuscript. As it stands now, it reads poor and is difficult to follow. It should be clear and more general.
3. Please cite the reference for serum lactate level measurement from arterial blood gas analyses.
4. There are no details mentioned in the figure legends. I found the text written in the figure very difficult to read and interpret. I think it is better to move all the details from the figures to figure legends.
5. The text written in the colored boxes in Fig.2 flow charts is very difficult to read. Please replace with the better resolution image.

---

## Round 0.2 · accepted · Accept

In my view, all the issues pointed by both reviewers were adequately addressed and the manuscript was amended accordingly. Therefore, I am please to accept the revised version of your manuscript.